# Micro-Milling of Additively Manufactured Al-Si-Mg Aluminum Alloys

**DOI:** 10.3390/ma17112668

**Published:** 2024-06-01

**Authors:** Qiongyi He, Xiaochong Kang, Xian Wu

**Affiliations:** 1School of Mechanical Engineering, Tianjin University of Technology and Education, Tianjin 300350, China; hqyyhy@vip.163.com; 2Jimei Industrial College, Xiamen 361022, China; kxc215@163.com; 3College of Mechanical Engineering and Automation, Huaqiao University, Xiamen 361021, China

**Keywords:** additively manufactured aluminum alloys, processability, burr formation

## Abstract

Additively manufactured aluminum alloy parts attract extensive applications in various felids. To study the machinability of additively manufactured aluminum alloys, micro-milling experiments were conducted on the additively manufactured AlSi7Mg and AlSi10Mg. By comparing the machinability of Al-Si-Mg aluminum alloys with different Si content, the results show that due to the higher hardness of the AlSi10Mg, the cutting forces are higher than the AlSi7Mg by about 11.8% on average. Due to the increased Si content in additively manufactured Al-Si-Mg aluminum alloys, the surface roughness of AlSi10Mg is 26.9% higher than AlSi7Mg on average. The burr morphology of additively manufactured aluminum alloys in micro-milling can be divided into fence shape and branch shape, which are, respectively, formed by the plastic lateral flow and unseparated chips. The up-milling edge exhibits a greater burr width than the down-milling edge. Due to the better plasticity of AlSi7Mg, the burr width of the down-milling edge is 28.1% larger, and the burr width of the up-milling edge is 10.1% larger than the AlSi10Mg. This research can provide a guideline for the post-machining of additively manufactured aluminum alloys.

## 1. Introduction

With the rapid development of lightweight design in various fields, such as aerospace, transportation, and automotive industries, more and more integrated structural components are being adopted. Lightweight manufacturing is gradually becoming a trend and greatly promotes the application of lightweight alloys in these fields. Aluminum alloy materials have excellent properties, such as low weight, good specific strength and stiffness, and higher plasticity, making them the preferred alloy for achieving a structural lightweight design [1,2,3]. The widespread application of aluminum alloy has promoted the rapid development of aluminum alloy materials and its machining technology. Traditional aluminum alloy components are mainly prepared by melting, casting, and forging, requiring a lot of time and energy. At present, with the continuous improvement of high-end equipment technology, higher requirements are put forward for the manufacturing technology of aluminum alloy components with complex and precise structures. This not only requires the manufacturing process to be efficient and precise but also to have high flexibility and adaptability requirements for its production and manufacturing process.

Additive manufacturing technology, also known as 3D printing technology, can directly manufacture parts layer-by-layer based on a digital model of the parts. Additive manufacturing technology can rapidly manufacture complex components as a whole. It has been widely applied in many fields, such as aerospace, biomedical, and rail transit [4,5,6]. With the rapid development of metal additive manufacturing technology, many researchers have reported on the additive manufacturing of aluminum alloys. Li et al. achieved the additive manufacturing of high-strength aluminum alloys through the laser powder bed fusion (LPBF) process [7]. Liu et al. prepared high-strength aluminum alloys with strengths exceeding 600 MPa using a pulse laser arc hybrid process [8]. At present, aluminum alloy additive manufacturing technology has been applied in industrial production to achieve the complex structural customization of products and shorten market response time [9,10,11].

Although metal additive manufacturing technology presents advantages in terms of flexibility and efficiency, due to limitations in machining principles, the aluminum alloy parts prepared by additive manufacturing often cannot meet machining requirements regarding simultaneous machining accuracy and surface quality. Additionally, it usually requires subsequent post-processing, such as mechanical machining and polishing [12,13,14]. Compared to metal materials prepared by casting and forging, metal materials prepared through additive manufacturing have significant differences in microstructure and mechanical properties, such as hardness and yield strength [15,16,17], which cause corresponding changes in their machinability in the mechanical machining process. Wu et al. found significant differences in the cutting machinability of additively manufactured titanium alloy compared to the forged titanium alloy. They pointed out that the machining parameters must be adjusted for the additively manufactured blank [18]. Randolph et al. performed single-point turning on additively manufactured A205 aluminum alloys; the results indicated that additively manufactured A205 material presented good ultra-precision turning performance [14]. Segebad et al. investigated the influence of build-up direction on chip formation in the orthogonal cutting of additively manufactured AlSi10Mg [19]. Tan et al. studied the ultrasonic elliptical vibration-assisted cutting of selective laser melting (SLM) additively manufactured AlSi10Mg alloy and found it can suppress surface defects and improve surface quality [20]. Although many researchers have conducted extensive studies on the cutting machinability of various additively manufactured metal materials [21,22,23], there are few reports on the effect of the Si phase in additively manufactured aluminum alloy on micro-milling machinability in the existing literature.

This paper focuses on the subsequent post-milling of the additively manufactured Al-Si-Mg aluminum alloys. Micro-milling experiments were conducted on AlSi7Mg and AlSi10Mg using a micro end mill. The milling force, surface roughness, and burr formation have been analyzed. The cutting performance of additively manufactured aluminum alloys with different Si contents was studied. The results can provide theoretical guidance for industrial applications in the production of aluminum alloy parts by additive manufacturing.

## 2. Experimental Procedures

In this work, micro-milling research was performed on a five-axis machine center (JDGR 200T, Beijing Jingdiao Group, Beijing, China), which is specially designed for the machining of micro components, as shown in Figure 1. This machine can provide a maximum spindle rotating speed of 24,000 rpm and a repeated position accuracy of 2 μm. The tool used in the experiment was a micro end mill with two flutes made of fine-grained cemented carbide (MSE 230, NS Tool Co., Ltd., Tokyo, Japan), which is specially designed for micro machining. Tool diameter was measured to 1 mm, the edge length was 2.5 mm, handle diameter was 4 mm, and helix angle was 30°; tool structure parameters are shown in Figure 2.

When reducing weight is the main goal, additively manufactured aluminum alloys are a common choice for aerospace and high-performance racing applications. The Al-Si-Mg alloys present excellent laser machining performance and attract much attention in the additive manufacturing field of selective laser melting (SLM) processes. Hence, the Al-Si-Mg alloys have been selected to study the micro-milling machinability. The used workpiece materials in this work were AlSi7Mg and AlSi10Mg aluminum alloys prepared by the SLM process, produced by Falcon Tech Co., Ltd., Wuxi, China. In SLM, the laser power was 340 kW, the scanning speed was 1400 m/s, the scanning interval was 90 μm, and layer thickness was 30 μm. The printing orientation deflects a 67° angle from the length direction of the workpiece. The workpiece was cut into the suitable size of 35 × 9 × 3 mm by wire electric discharge machining (WEDM) and pre-machined by conventional milling. The chemical compositions and mechanical properties along the horizontal direction are listed in Table 1 and Table 2, based on the previous literature [24,25]. Table 2 shows that the additively manufactured aluminum alloys present different mechanical properties due to different Si contents. In comparison, the yield strength of AlSi10Mg aluminum alloy is slightly lower in yield strength but slightly higher in material hardness.

In this work, straight groove micro-milling experiments were carried out on the top surface of the SLM samples with a water-based cutting fluid. The micro-milling machinability of two different alloy materials made by the SLM process was compared through single-factor experiments. Based on the previous preliminary experiments, the processing parameters used are listed in Table 3. Based on the fixed machining parameters *n* = 14,331 rpm, *f_z_
*= 2 μm/Z, and *a_p_* = 20 μm, the spindle speed varied in the range of (9554~23,885) rpm, the corresponding cutting velocity was (30~75) m/min, the feed rate was selected to vary in the scope of (1~4) μm/Z, and the variation range of the milling depth was (20~50) μm. During micro-milling experiments, the cutting force in the micro-milling process was measured using a dynameter (9257B, Kistler, Winterthur, Switzerland) with a sampling frequency of 20 kHz. The cutting force data were then analyzed by Dyno Ware software to obtain the cutting force results. After micro-milling, the burr width was tested using the optical microscope (VHX-1000, Keyence, Osaka, Japan), and the surface roughness Ra was inspected following the feed direction at the middle position of the machined groove using a white light interferometer (NewView 7300, Zygo, Middlefield, CT, USA).

## 3. Results and Discussions

### 3.1. The Cutting Force Comparison

In micro-milling experiments, the milling force signal in the machining process was recorded using a dynameter, and the difference value between the peaks and valleys in the milling force signal was used as the milling force results. The changes in the milling force of two additively manufactured aluminum alloys under different machining parameters are shown in Figure 3. The recorded milling force includes three cutting force components perpendicular to each other: Fx is the main cutting force, Fy is the feed force, and Fz is the axial force. Through comparison, we found that the main cutting force component was the largest, followed by the feed force component, and the axial force component was far less than the other two force components.

As the spindle speed increased from 9554 to 23,885 rpm, the milling forces of two additively manufactured aluminum alloys showed a slightly increase trend. Among them, three cutting force components of AlSi7Mg increased from 0.52 N, 0.36 N, and 0.13 N to 0.84 N, 0.49 N, and 0.23 N, respectively, while three cutting force components of AlSi10Mg increased from 0.56 N, 0.38 N, and 0.23 N to 0.92 N, 0.63 N, and 0.35 N, respectively. The increase in the main component of the milling force was more significant than the axial component. Due to the good plasticity of aluminum alloys, it is easy for them to adhere to tool surfaces and form a built-up edge. Within the cutting speed range in this experiment, as the cutting speed increases, more built-up edges adhere to the tool surface, leading to a gradual increase in the milling force. It is necessary to exceed this cutting velocity scope to obtain a smaller cutting force. As the feed rate and milling depth increase, three cutting force components of two additively manufactured aluminum alloys also gradually increase. This is because the increase in these two milling parameters directly results in an increase in the cutting area during the machining process, an increase in the required cutting power, and the corresponding cutting forces also increase.

Comparing three cutting force components of two additively manufactured aluminum alloys with different Si contents, it was found that the cutting force components of AlSi10Mg during micro-milling were relatively higher than the AlSi7Mg alloy. By counting three cutting force components of two additive manufactured aluminum alloys under different machining parameters and calculating the average value, we found that the main cutting force component Fx, feed force component Fy, and axial force component Fz of AlSi10Mg were, respectively, 21.9%, 10.4%, and 30.7% higher than that of AlSi7Mg. It can be inferred from the mechanical properties of two additively manufactured aluminum alloys that the hardness of AlSi10Mg material is slightly higher than that of AlSi7Mg, which is the main reason for the increased milling force.

The resultant forces were also calculated (except the cutting force components). The resultant force comparison of two additively manufactured aluminum alloys under different machining parameters is shown in Figure 4. From the results, it was found that the resultant forces of AlSi10Mg were also larger than that of AlSi7Mg. The average resultant forces were about 11.8% larger with the increased Si content in AlSi10Mg aluminum alloys.

### 3.2. The Surface Roughness Comparison

In industrial production, the surface roughness is commonly used to reflect the surface quality of the mechanically processed workpieces. The main measurement methods of surface roughness include the contact and non-contact methods. Due to the fact that micro-groove machined parts by micro-milling are narrow, with only a width of 1 mm, the probe in the contact measurement method cannot enter the micro-grooves to measure the surface roughness. Therefore, in this experiment, the non-contact method was employed to inspect the surface roughness Ra of the milled surface by a white light interferometer, as shown in Figure 5. Surface roughness, Ra, was measured following the feed direction at the middle position of the machined surface and was repeated three times at different positions; their average value is in the experimental results.

The surface roughness of two additively manufactured aluminum alloys with different machining parameters is shown in Figure 6. From the results, with the change in machining parameters, the variation in the surface roughness trend of two additively manufactured aluminum alloys was nearly consistent. As the spindle speed increased, the inspected surface roughness showed a gradually decreasing trend. Higher cutting speeds can achieve a good machining surface quality. When the feed rate rises from 1 to 2 μm/Z, the surface roughness slightly decreases first; when the feed rate rises from 2 to 4 μm/Z, there is a significant upward trend in surface roughness. It can be seen that the surface roughness is also influenced by the minimum cutting thickness and size effect, and when the cutting thickness is too small, it forms a larger machining surface roughness. As the milling depth rises, the surface roughness slightly increases, and the effect of milling depth on the machining surface roughness seems relatively smaller than the other two machining parameters.

Comparing two additively manufactured aluminum alloy materials, it was found that the inspected surface roughness of the AlSi7Mg was relatively smaller than that of the AlSi10Mg. By calculating the averaged surface roughness of two additively manufactured aluminum alloys under different machining parameters, it was found that the surface roughness of the AlSi10Mg was 26.9% higher than that of the AlSi7Mg. This result indicates that an increase in Si content in the additively manufactured aluminum alloy will, to some extent, deteriorate the machined surface quality. From Figure 5, it was observed that there were some micro pits on the surface morphology and profile of the micro-milled surface. During SLM, the Si phase in the aluminum alloy is easy to oxidize; then, the hard SiO_2_ particle formed can fall off in the cutting process and is unfavorable to obtain a good surface quality. Hence, the increase in the Si phase in the additively manufactured aluminum alloy results in higher surface roughness in micro-milling.

### 3.3. The Burr Formation Comparison

The burr located on the top edge is a common trouble that affects the machining quality and the subsequent assembly. The top burr morphologies observed during the micro-milling of the additively manufactured aluminum alloys are shown in Figure 7. From the results, it was found that the top burrs exist on the top edges of both the top and bottom milling edges. The top burr can be classified into fence shape or branch shape based on their typical morphological characteristics. The fence shape burrs are continuous, with relatively uniform width and height, similar to continuous fences erected at the machined edge. The branch shape burrs are intermittent, and their orientation is relatively consistent with the obliques to the feed direction. They are connected to the top edge of the micro-grooves just like branches, and this burr width varies greatly along the feed direction. The micro-milling process is an intermittent cutting process, and the formation of the branch shape burrs is not very stable in this intermittent cutting process.

During the cutting process, the burrs in micro-milling are formed by two main reasons. When the cutting thickness is very small, even lower than the minimum cutting thickness, the machined material under the cutting edge does not flow along the tool rake surface but has plastically lateral flow due to the dramatic squeezing and plowing. This material plastic lateral flow behavior usually forms burrs standing upright to the machined edge, which is the fence-shaped burr, as shown in Figure 8. Additionally, some chips in the cutting process do not completely separate from the workpiece due to insufficient motivation or edge obstruction. These unseparated chips during the cutting process also form large burrs on the machined edge.

The height and width of burrs are commonly used indicators to measure burr size. Among these two indicators, the burr width is relatively easier to measure. Therefore, in this experiment, the burr width is employed to assess burr size under different parameters. As exhibited in Figure 9, the burr width of the up-milling and down-milling edges is measured at different positions of the micro-milled groove; their averaged value is taken as the experimental results.

The inspected burr width results with different machining parameters are shown in Figure 10. Comparing the burr size located on the up-milling and down-milling edges, the burr size on the down-milling edge are, to varying degrees, greater than those on the up-milling edge for both additively manufactured AlSi7Mg and AlSi10Mg. During the micro-milling process, with the tool rotating, the cutting edge cuts in at the up-milling edge, and its instantaneous cutting thickness gradually raises from zero to the maximum value, then cuts out at the down-milling edge, and the instantaneous cutting thickness gradually downwards to zero again, as exhibited in Figure 11. At the up-milling edge, due to the small instantaneous cutting thickness, the plastic lateral flow materials will generate fence burrs. The cutting edge pushes the unseparated chips along the cutting velocity direction from the machined edge to the unmachined area. These unseparated chips do not leave the machined edge, resulting in a smaller burr width. However, during the cutting process at the down-milling edge, the cutting edge will push some unseparated chips along the cutting velocity direction from the unmachined area to the machined edge. This behavior generates the branch-shaped burrs on the down-milling edge in each cutting pass. With the continuous feed of the tool, the branch-shaped burr continuously generates, obtaining a larger burr width.

From the results in Figure 10, the burr width of both the additively manufactured AlSi7Mg and AlSi10Mg alloys shows a consistent trend with the variation in machining parameters. As the spindle speed increases, the burr width of both materials in up-milling and down-milling edges gradually decreases. Since the feed rate increases from 1 to 3 μm/Z, the burr width gradually decreases first; when the feed rate becomes greater than 3 μm/Z, the burr width tends to stabilize and shows a slight upward trend. It can be observed that when the feed rate is too low, the cutting process is affected by the minimum cutting thickness and size effect, resulting in a larger burr width during the machining process. As the milling depth increases, the burr width of two materials in the up and down-milling edges generally shows an upward trend. With the milling depth increasing from 20 to 40 μm, the burr width is relatively stable. But since the milling depth rises from 40 to 50 μm, the burr width begins to show a relatively significant increase. When the milling depth is large, it is equivalent to a larger cutting width in the orthogonal cutting process. In this case, the larger and longer chips will be formed during the cutting process, and some of the remaining chips will be pushed and adhered to the workpiece edge, forming relatively large burr sizes.

Comparing the additively manufactured AlSi7Mg and AlSi10Mg aluminum alloys, it is found that the burr width of AlSi7Mg aluminum alloy material is relatively larger than that of AlSi10Mg. By calculating the average burr width of two materials under different processing parameters, it is indicated that the burr width on the up-milling edge of the AlSi7Mg material is 28.1% larger, and the burr width on the down-milling edge is 10.1% higher than that of the AlSi10Mg. By comparing the mechanical properties of the additively manufactured AlSi7Mg and AlSi10Mg aluminum alloys, it was found that the elongation of the AlSi7Mg material is relatively higher than that of the AlSi10Mg; this indicates that its plasticity is relatively good and the burr size generated during micro-milling is larger.

## 4. Conclusions

In this paper, micro-milling experiments were performed on the additively manufactured AlSi7Mg and AlSi10Mg alloys with different Si contents. The effect of Si contents in the additively manufactured aluminum alloys on its micro-milling machinability was studied in terms of cutting force, surface roughness, and burr width. Based on the results, the conclusions can be drawn as follows:The cutting force of the AlSi7Mg and AlSi10Mg alloys increases with the increase in cutting parameters. Because of the higher material hardness of AlSi10Mg, the cutting force component Fx is 21.9% higher, the feed force Fy is 10.4% higher, and the axial force Fz is 30.7% higher than AlSi7Mg, on average.Surface roughness in the micro-milling of AlSi7Mg and AlSi10Mg alloys decreases with rising spindle speed and increases with rising feed rate and milling depth. The increased Si content of the additively manufactured aluminum alloys increases surface roughness. The surface roughness of AlSi10Mg is, on average, 26.9% higher than that of AlSi7Mg.Burr morphology in the micro-milling of additively manufactured aluminum alloys can be divided into fence and branch shapes, and the burr width on the down-milling edge is greater than up milling edge. In terms of machining parameters, the burr width gradually decreases with increased spindle speed. Because of the minimum cutting thickness and size effect, the burr width first decreases and then slightly increases with the feed rate. As the milling depth increases, the burr width shows a gradual increasing trend. Due to the relatively better plasticity of the AlSi7Mg, the generated burr width during the machining process is 28.1% larger on the up-milling edge and 10.1% larger on the down-milling edge than that of the AlSi10Mg.

## Figures and Tables

**Figure 1 materials-17-02668-f001:**
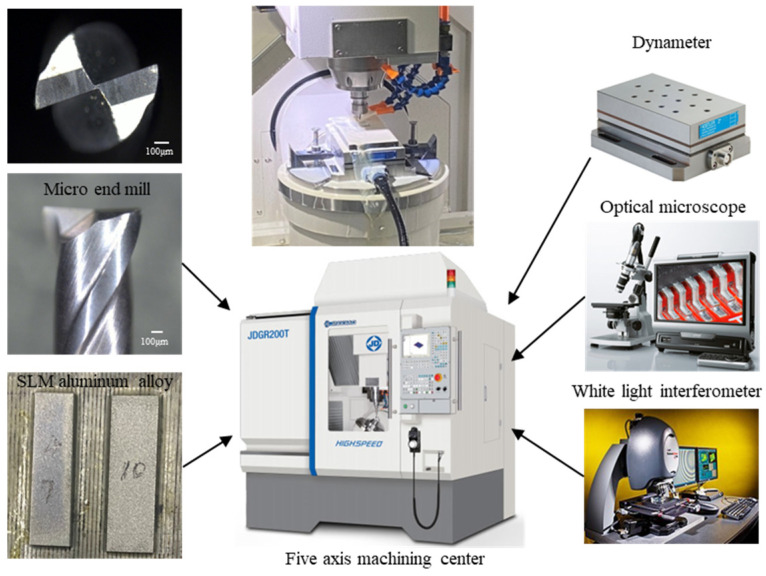
Micro-milling experiments.

**Figure 2 materials-17-02668-f002:**
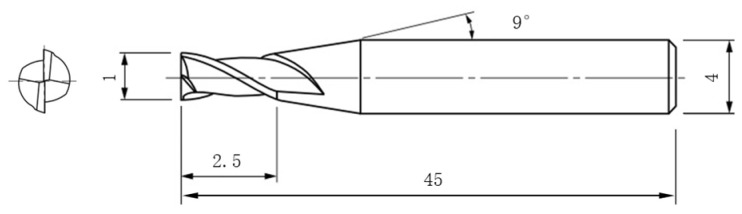
Tool structure parameters of micro end mill (mm).

**Figure 3 materials-17-02668-f003:**
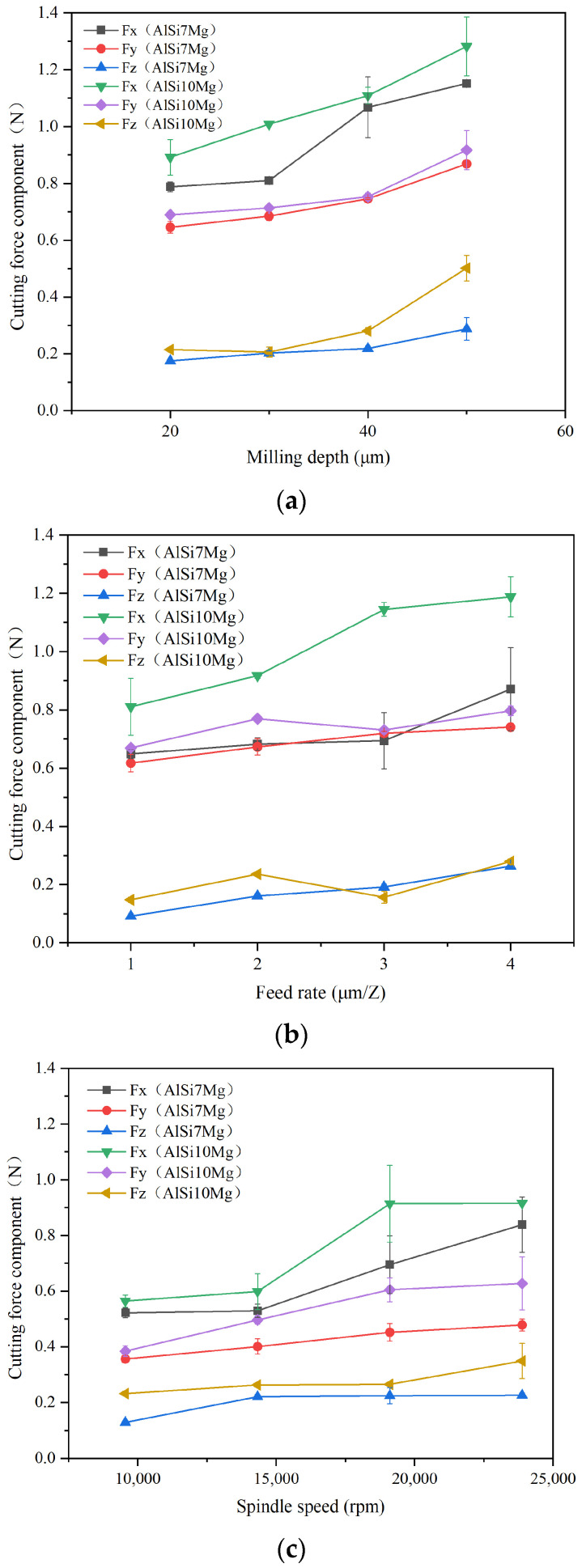
The milling force components comparison between additively manufactured AlSi7Mg and AlSi10Mg alloys. (**a**) The milling force components under different spindle speeds. (**b**) Milling force components under different feed rates. (**c**) Milling force components under different milling depths.

**Figure 4 materials-17-02668-f004:**
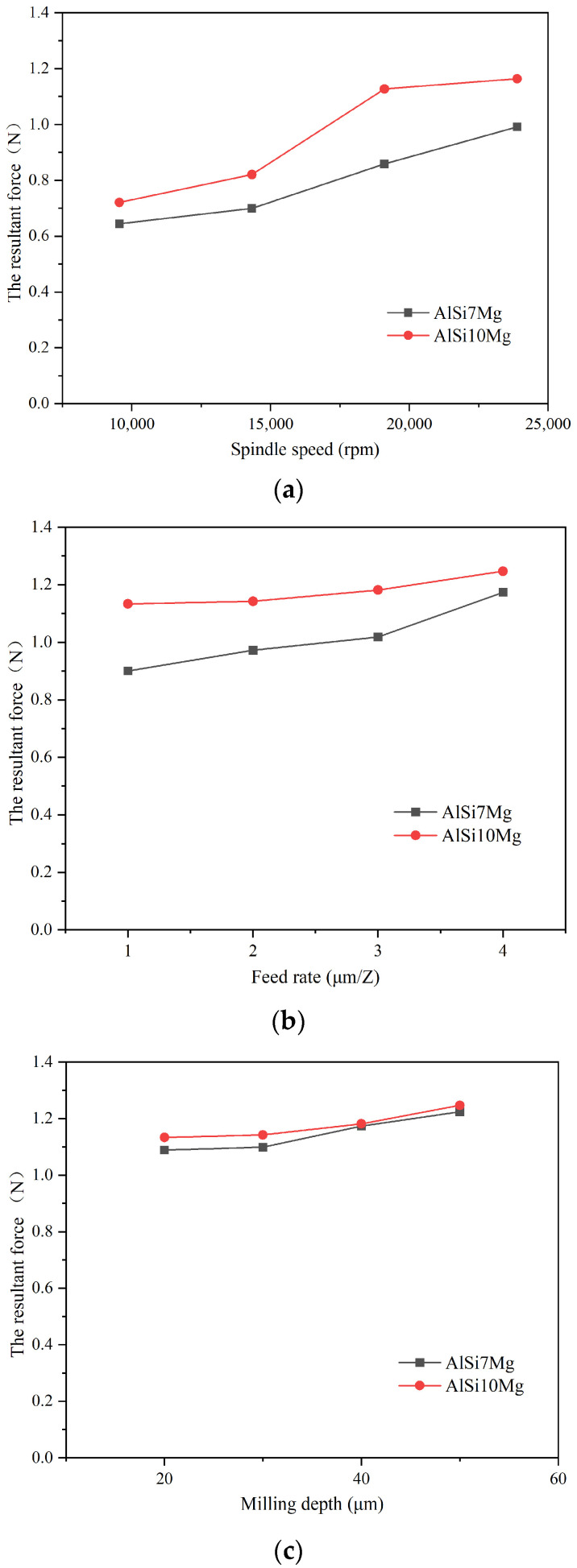
The resultant force comparison between additively manufactured AlSi7Mg and AlSi10Mg alloys. (**a**) The resultant force under different spindle speeds. (**b**) The resultant force under different feed rates. (**c**) The resultant force under different milling depths.

**Figure 5 materials-17-02668-f005:**
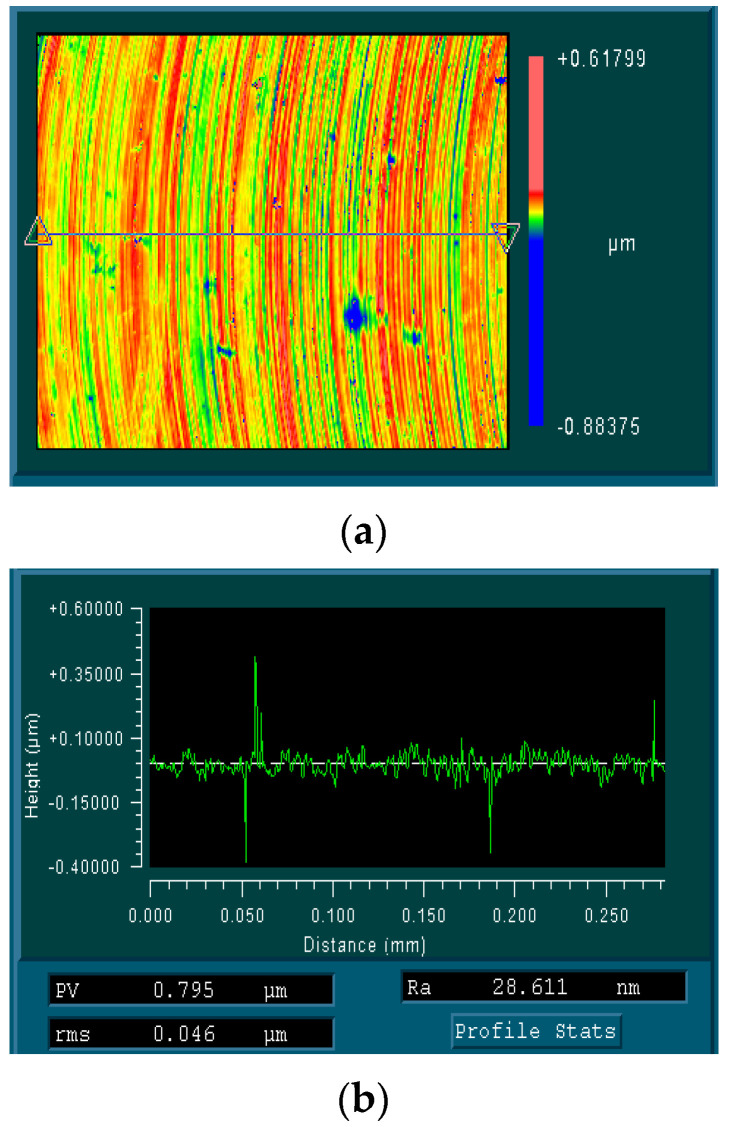
Surface roughness measurement (material: additively manufactured AlSi10Mg, parameters: *n* = 14,331 rpm, *f_z_* = 2 μm/Z and *a_p_
*= 30 μm). (**a**) Surface morphology and roughness measurement direction. (**b**) Surface profile and roughness Ra.

**Figure 6 materials-17-02668-f006:**
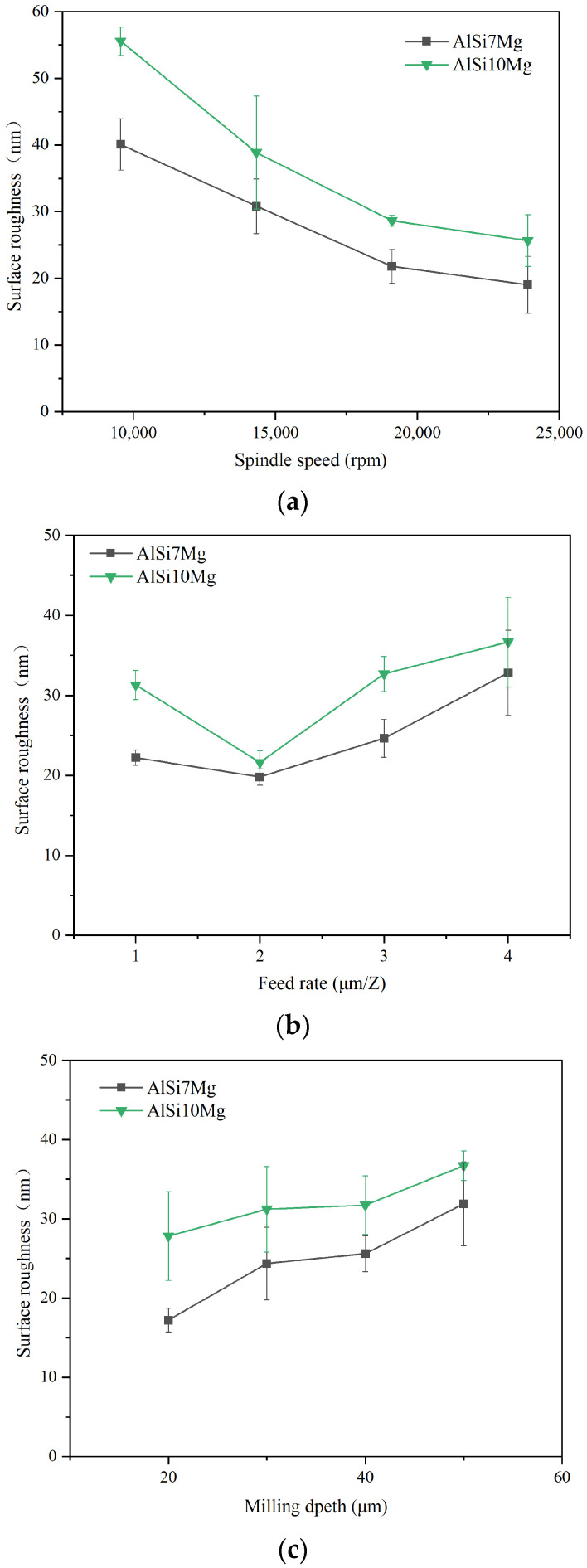
The surface roughness comparison between additively manufactured AlSi7Mg and AlSi10Mg alloys. (**a**) Surface roughness under different spindle speeds. (**b**) Surface roughness under different feed rates. (**c**) Surface roughness under different milling depths.

**Figure 7 materials-17-02668-f007:**
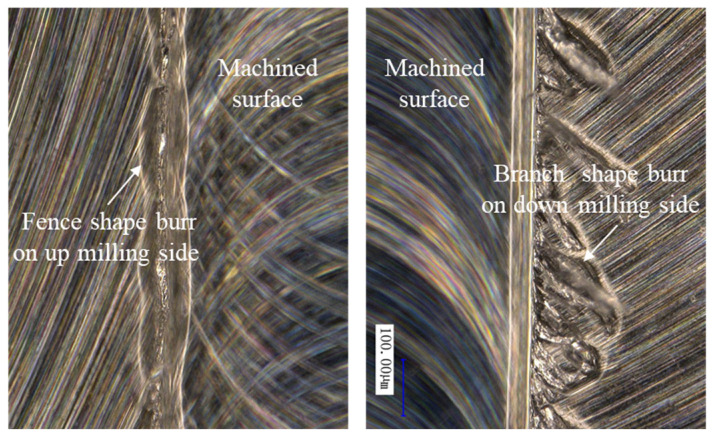
Burr morphology characteristics.

**Figure 8 materials-17-02668-f008:**
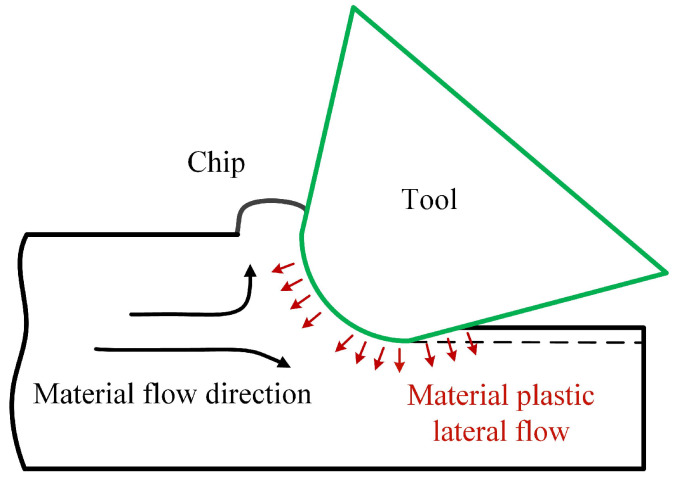
Material plastic lateral flow under the cutting edge.

**Figure 9 materials-17-02668-f009:**
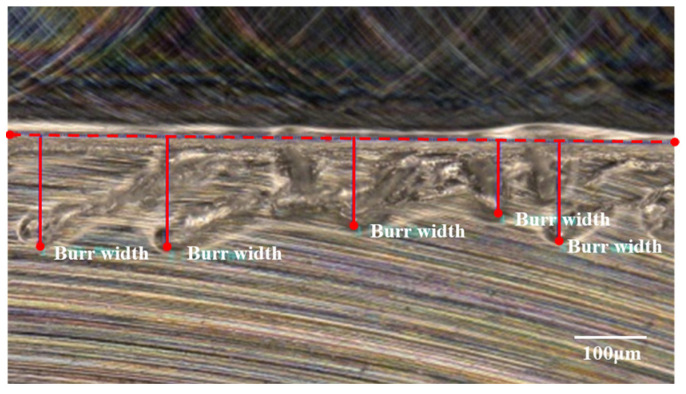
Burr width measurement.

**Figure 10 materials-17-02668-f010:**
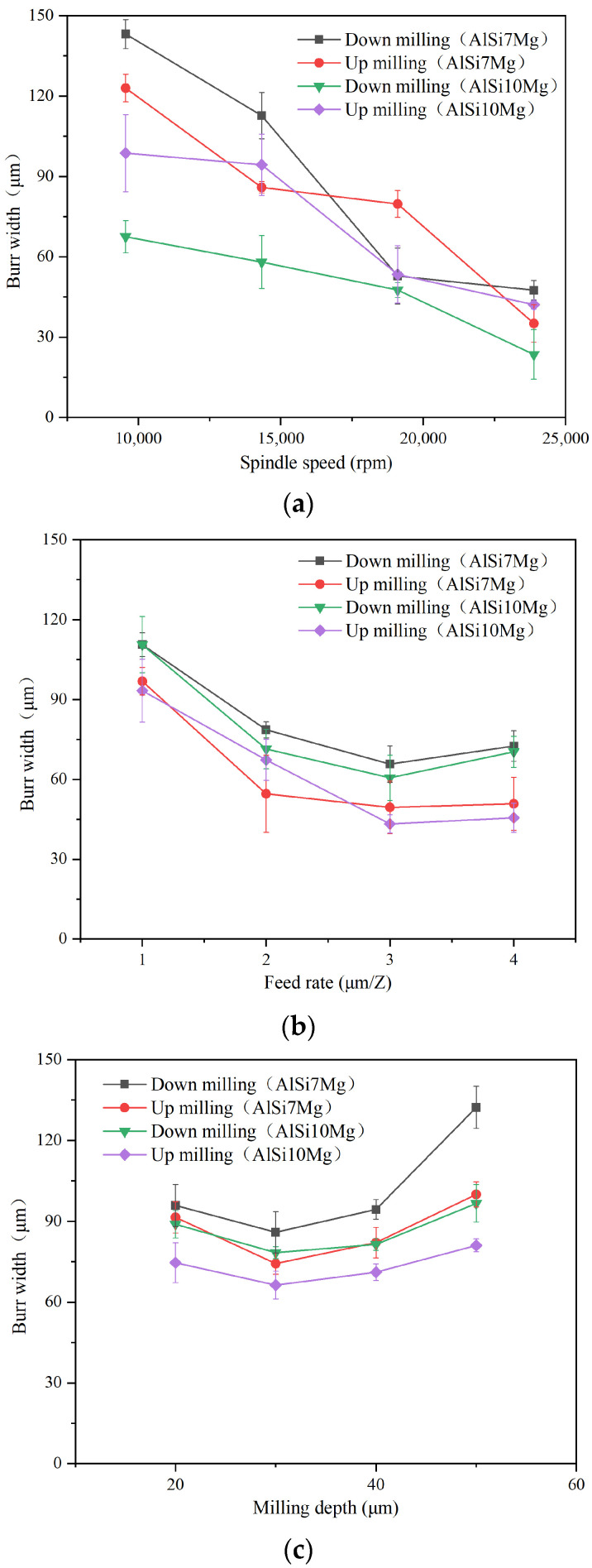
Burr width comparison between additively manufactured AlSi7Mg and AlSi10Mg alloys. (**a**) Burr width under different spindle speeds. (**b**) Burr width under different feed rates. (**c**) Burr width under different milling depths.

**Figure 11 materials-17-02668-f011:**
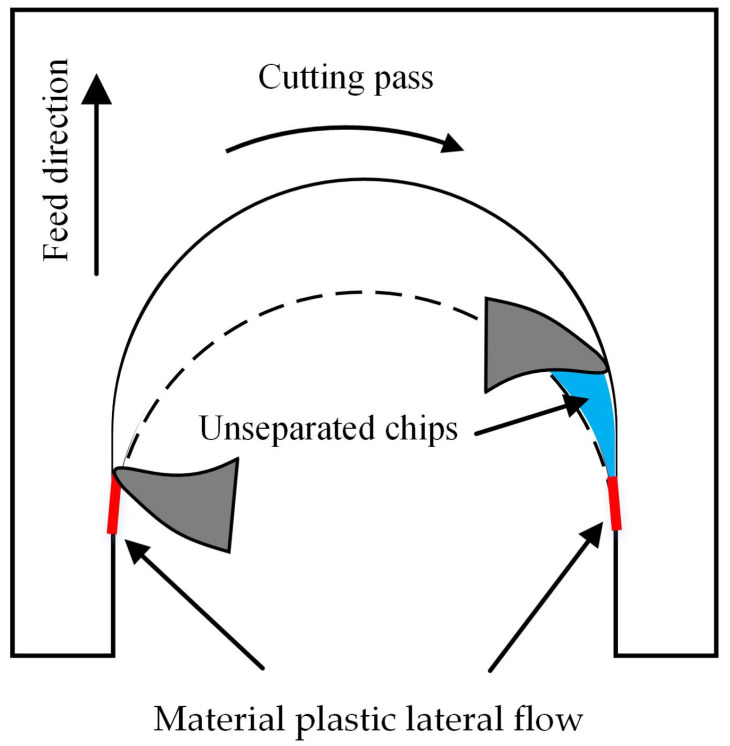
Burr formation in micro-milling.

**Table 1 materials-17-02668-t001:** The chemical compositions of the additively manufactured aluminum alloys.

Material	Si wt%	Mg wt%	Fe wt%	Zn wt%	Ti wt%	Ni wt%	Al wt%
AlSi7Mg	7	0.36	0.1	0.016	0.006	0.004	Bal
AlSi10Mg	10	0.36	0.1	0.016	0.006	0.004	Bal

**Table 2 materials-17-02668-t002:** The mechanical properties of the additively manufactured aluminum alloys.

Material	Yield Strength (MPa)	Elongation (%)	Hardness (HV)
AlSi7Mg	299	14.3	110.5
AlSi10Mg	280	8.1	120.7

**Table 3 materials-17-02668-t003:** Processing parameters.

Parameter Name and Unit	Value
Spindle speed *n*/rpm	9554, 14,331, 19,108, 23,885
Feed rate *f_z_* μm/Z	1, 2, 3, 4
Milling depth *a_p_* μm	20, 30, 40, 50

## Data Availability

The data that support this study are available within this article.

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
