# Peer review of "Micro-Milling of Additively Manufactured Al-Si-Mg Aluminum Alloys"

_materials, 2024, doi:10.3390/ma17112668_

Round 1

Reviewer 1 Report

Comments and Suggestions for Authors

In this work, experiments were carried out on microgrinding of aluminum alloys AlSi7Mg and AlSi10Mg obtained by additive method.

1) “To study the processability of additive manufactured aluminum alloys 12 Al-Si-Mg, micro milling experiments were conducted on the additive manufactured AlSi7Mg and 13 AlSi10Mg.” As far as I understand, the main advantage of alloys produced by additive manufacturing is precisely the absence of the need for post-processing. What is the point of printing an alloy and then further mechanically processing it? It’s easier to take an alloy produced by casting.

2) The authors do not explain the choice of materials for the study.

3) “Randolph et al. performed single point turning on additive manufactured A205 alumi-66 num, the results indicated that additive manufactured A205 is machinable [14].” Are there any alloys that cannot be machined?

4) Please add size mark to the sample picture in picture 1.

5) “Yield strength (MPa)Elongation (%)Hardness (HV)” please add confidence intervals for these values. How were these values obtained? What samples and how many of them were tested?

6) While reading the article, the question arises: why do you constantly indicate that these alloys are “additive manufactured”. It seems that somewhere in your work you are planning to compare with alloys obtained by a different method.

7) “Figure 5 Surface roughness measurement” for which alloy?

8) “of the additive manufactured AlSi10Mg is % higher than that of the” what amount of %?

9) “by the fall off of hard particle of Si phase.” please explain what particles you are talking about.

10) “Figure 7 Burr morphology characteristics” “Figure 9 Burr width measurement” what is shown in the figure? Size mark? Please add more information

in general, the article seems more applied than scientific work

Author Response

Response to Reviewer 1 Comments

We would like to thank the reviewer for giving us the constructive suggestions which would help us to improve the paper quality. Here we submit a new version of our manuscript which has been revised according to the suggestions. All the revisions have been highlighted in the manuscript. We hope that our revised manuscript and response to reviewers submitted here can meet the requirement of the journal Materials. The following is a point-by-point response to the Reviewer 1 comments:

Point 1: “To study the processability of additive manufactured aluminum alloys 12 Al-Si-Mg, micro milling experiments were conducted on the additive manufactured AlSi7Mg and 13 AlSi10Mg.” As far as I understand, the main advantage of alloys produced by additive manufacturing is precisely the absence of the need for post-processing. What is the point of printing an alloy and then further mechanically processing it? It’s easier to take an alloy produced by casting.

Response 1: Thank you for the comments. At present, the additive manufacturing technology can achieve the rapid manufacturing of complex components as a whole. It has been widely applied in many fields, such as aerospace, biomedical, and rail transit. Although the metal additive manufacturing technology presents the advantages in terms of flexibility and efficiency, however, however, the parts prepared by the additive manufacturing often cannot fully meet the machining requirements in terms of machining accuracy and surface quality at once. So, it usually requires the subsequent post-processing, such as mechanical machining or polishing.

Point 2: The authors do not explain the choice of materials for the study.

Response 2: Thank you for the comments. When reducing weight is the main goal, the additive manufactured aluminum alloys are a common choice for aerospace and high-performance racing applications. The Al-Si-Mg alloys present excellent laser machining performance, and attract many attentions in the additive manufacturing flied of selective laser melting process. Hence, the Al-Si-Mg alloy materials have been selected to study the micro milling machinability.

Point 3: “Randolph et al. performed single point turning on additive manufactured A205 alumi-66 num, the results indicated that additive manufactured A205 is machinable [14].” Are there any alloys that cannot be machined?

Response 3: Thank you for the comments. Your suggestion has been adopted, this description has been corrected. Randolph et al. performed single point turning on additive manufactured A205 aluminum alloys, the results indicated that additive manufactured A205 material presented good ultra-precision turning performance.

Point 4: Please add size mark to the sample picture in picture 1.

Response 4: Thank you for the comments. Your suggestion has been adopted, this picture has been corrected.

Point 5: “Yield strength (MPa)Elongation (%)Hardness (HV)” please add confidence intervals for these values. How were these values obtained? What samples and how many of them were tested?

Response 5: Thank you for the comments. The mechanical properties of the additive manufactured aluminum alloys are referred from the previous literatures.

Point 6: While reading the article, the question arises: why do you constantly indicate that these alloys are “additive manufactured”. It seems that somewhere in your work you are planning to compare with alloys obtained by a different method.

Response 6: Thank you for the comments. The frequently occurred word “additive manufactured” has been revised.

Point 7: “Figure 5 Surface roughness measurement” for which alloy?

Response 7: Thank you for the comments. Surface roughness measurement in Figure 5 is obtained from the additive manufactured AlSi10Mg with the machining parameters of n = 14331 rpm, fz =2 μm/Z and ap=30 μm.

Point 8: “of the additive manufactured AlSi10Mg is % higher than that of the” what amount of %?

Response 8: Thank you for the comments. The percentage in this paper is counting of all data (cutting force, surface roughness and burr width), and then calculating the average value.

Point 9: “by the fall off of hard particle of Si phase.” please explain what particles you are talking about.

Response 9: Thank you for the comments. This description has been revised. During SLM, the Si phase in aluminum alloy is easy to oxidation, the formed hard particle SiO2 is possible to fall off in the cutting process, and unfavorable to obtain good surface quality.

Point 10: “Figure 7 Burr morphology characteristics” “Figure 9 Burr width measurement” what is shown in the figure? Size mark? Please add more information.

Response 10: Thank you for the comments. In Figure 7, the burr morphologies have been shown. From this picture, based on their typical morphological characteristics, the top burrs have been classified into three types, fence shape and branch shape. And then, the formation mechanisms of three types of top burrs have been analyzed. In Figure 9, the burr width was measured to analyze the effect of machining parameters on burr size. The size mark has been added in the picture.

Reviewer 2 Report

Comments and Suggestions for Authors

1. At the end of the Introduction section, the research focus is written. In addition, it would be desirable to explicitly write the scientific contribution and scientific hypotheses.

2. Section 2 requires more significant corrections. First, it should be written explicitly what is new in applied methods and/or experimental research. In this way, the authors will emphasize the originality of the research.

3. In section 2, the authors list the selected materials, equipment, milling parameters, etc. However, the authors do not state how the selection was made. For example, how are the selected milling parameters? Why are these parameters representative for research?

4. How is the cutting tool selected? Why is it representative of this research?

5. For both workpiece materials, it would be desirable to show a detailed chemical composition. It would also be desirable to show the physical and thermal properties of the workpieces material.

6. It is also not the clearest use of the term "micro milling". In some places in the manuscript it is "micro milling" and in others "milling". Also, where is the boundary between "micro milling" and "milling"? Why is it emphasized that this is micro milling?

7. What experimental design was used? Why?

8. The obtained results for all combinations of input parameters could be tabulated.

9. What is the initial roughness for each experiment?

10. Why was the surface roughness measured only in the feed direction?

11. The effects of milling parameters on milling force components, surface roughness and burr width were considered separately. Why? Is there an interaction between the parameters?

12. What is the uncertainty of the obtained results?

13. Discussion section is missing. This section must be inserted. At present, the results have been narrowly debated. It is not enough to state whether something is bigger or smaller, whether something is increasing or decreasing, etc. It is necessary to explain why this is so, the results should be scientifically discussed (influence of input parameters on output parameters.

14. The Conclusions section should be corrected and updated. Currently, the most important results are repeated in this section. In this section, the scientific contribution and possibilities of practical application should be highlighted. Limitations of the research and directions for future research should also be written in this section.

Author Response

Response to Reviewer 2 Comments

We would like to thank the reviewer for giving us the constructive suggestions which would help us to improve the paper quality. Here we submit a new version of our manuscript which has been revised according to the suggestions. All the revisions have been highlighted in the manuscript. We hope that our revised manuscript and response to reviewers submitted here can meet the requirement of the journal Materials. The following is a point-by-point response to the Reviewer 2 comments:

At the end of the Introduction section, the research focus is written. In addition, it would be desirable to explicitly write the scientific contribution and scientific hypotheses.

Response: Thank you for the comments. The research focus has been added at the end of the Introduction section. This paper mainly focuses on the subsequent post-milling of the additive manufactured Al-Si-Mg aluminum alloys. The results can provide the theoretical guidance for the industrial applications for production of aluminum alloy parts by additive manufacturing.

Section 2 requires more significant corrections. First, it should be written explicitly what is new in applied methods and/or experimental research. In this way, the authors will emphasize the originality of the research.

Response: Thank you for the comments. The detailed experimental methods have been revised based on your suggestions.

In section 2, the authors list the selected materials, equipment, milling parameters, etc. However, the authors do not state how the selection was made. For example, how are the selected milling parameters? Why are these parameters representative for research?

Response: Thank you for the comments. The milling parameters were selected based on our previous preliminary experiments, which by combining with the machine tool performance (maximum spindle rotating speed of 24000 rpm). The feed per tooth was selected to conclude the cases that larger and less than the cutting edge radius to reflect the size effect on the micro milling process.

How is the cutting tool selected? Why is it representative of this research?

Response: Thank you for the comments. The used tool in the work was micro end mill with two flute that made of fine grain cemented carbide, which is specially designed for micro machining. This tool is common applied in micro milled of metal materials, such as aluminum alloy and stainless steel.

For both workpiece materials, it would be desirable to show a detailed chemical composition. It would also be desirable to show the physical and thermal properties of the workpieces material.

Response: Thank you for the comments. The detailed chemical compositions of both workpiece materials have been added in the paper.

It is also not the clearest use of the term "micro milling". In some places in the manuscript it is "micro milling" and in others "milling". Also, where is the boundary between "micro milling" and "milling"? Why is it emphasized that this is micro milling?

Response: Thank you for the comments. In general, when the used tool diameter larger than 1mm, it is defined as conventional milling. And when tool diameter less than 1mm, it is defined as micro milling. In comparison, micro milling process is significantly affected by the size effect and minimum uncut chip thickness, so it is more difficult than conventional milling.

What experimental design was used? Why?

Response: Thank you for the comments. In this work, single factor experiment method was used to study the effect of machining parameters on the cutting force, surface roughness and burr formation.

The obtained results for all combinations of input parameters could be tabulated.

Response: Thank you for the comments. Your suggestion is very good. We considered that the tables and pictures of the obtained results would be duplicated, so we did not create the additional tables.

What is the initial roughness for each experiment?

Response: Thank you for the comments. Before micro milling experiment, the workpiece was cut into the suitable size of (35×9×3) mm by wire electric discharge machining (WEDM), and pre-machined by conventional milling, the intimal roughness of workpiece was about 1.6 μm.

Why was the surface roughness measured only in the feed direction?

Response: Thank you for the comments. This is the common measurement method for milled surface, because the tool mark texture in feed direction is regular, and closely related with the machining parameters. Surface roughness in feed direction can most accurately reflect the actual machining surface quality.

The effects of milling parameters on milling force components, surface roughness and burr width were considered separately. Why? Is there an interaction between the parameters?

Response: Thank you for the comments. Single factor experiment method was used in this work, so the effects of parameters on force components, surface roughness and burr width were considered separately. There is a certain interaction between the milling parameters, but it is restively low.

What is the uncertainty of the obtained results?

Response: Thank you for the comments. In this work, the measurement of cutting force, surface roughness and burr width have been repeated more than three times, and their average values were adopted as the results to avoid the uncertainty, in addition, the error bar also has been added for the results. 

Discussion section is missing. This section must be inserted. At present, the results have been narrowly debated. It is not enough to state whether something is bigger or smaller, whether something is increasing or decreasing, etc. It is necessary to explain why this is so, the results should be scientifically discussed (influence of input parameters on output parameters.

Response: Thank you for the comments. Your suggestion is very good. The discussion is important for a standardized academic paper. In this paper, we merged the results and discussion together without separating separate chapters. The detailed explanation and analysis have been presented after the experiment results.

The Conclusions section should be corrected and updated. Currently, the most important results are repeated in this section. In this section, the scientific contribution and possibilities of practical application should be highlighted. Limitations of the research and directions for future research should also be written in this section.

Response: Thank you for the comments. We have updated the conclusions based on your suggestions.

Reviewer 3 Report

Comments and Suggestions for Authors

It may be an interesting paper but the results are dealt with very roughly, which should be discussed in much more detail and with high precision. There are plenty of rooms to be improved and upgraded, in particular Fig. 3, Fig. 4, Fig. 6, and Fig. 11. I am not entirely sure how many measurements the authors carried out for each measurement, and apparently there are no error bars in these figures. 

Comments on the Quality of English Language

No comments.

Author Response

Response to Reviewer 3 Comments

We would like to thank the reviewer for giving us the constructive suggestions which would help us to improve the paper quality. Here we submit a new version of our manuscript which has been revised according to the suggestions. All the revisions have been highlighted in the manuscript. We hope that our revised manuscript and response to reviewers submitted here can meet the requirement of the journal Materials. The following is a point-by-point response to the Reviewer 3 comments:

It may be an interesting paper but the results are dealt with very roughly, which should be discussed in much more detail and with high precision. There are plenty of rooms to be improved and upgraded, in particular Fig. 3, Fig. 4, Fig. 6, and Fig. 11. I am not entirely sure how many measurements the authors carried out for each measurement, and apparently there are no error bars in these figures.

Response: Thank you for the comments. The cutting force and surface roughness measurements have been repeated for three times. The burr width measurements have been repeated for 5~10 times. Based on your suggestions, the error bars have been added in the pictures.

Reviewer 4 Report

Comments and Suggestions for Authors

The paper investigated the machineability of additively manufactured AlSi10Mg and AlSi7Mg with micro-milling. The hardness and surface roughness of the alloy with higher Si content is respectively approximately 12% and 27% higher. The burr morphology was classified to fence and branch shape. AlSi7Mg has better plasticity, leading to the burr width nearly 28% larger in down milling, and 10% larger in up milling in comparison with AlSi10Mg.
The authors can consider the below feedback. 
Format:
- Line 51: "600Mpa" please correct to "600 MPa"
- Abbreviations: LPBf and SLM are not explained. "Additive manufacturing" is usually abbreviated as "AM". Kindly check the rest.
- Line 62: "Wu et al. have found the significant differences in cutting machinability of additive manufactured titanium alloy compared to the forged titanium alloy, they have pointed out that the machining parameters need to be adjusted for the additive manufactured blank [18]." is a long sentence. Consider breaking it into two.
- Line 71: "From above, at present, many researchers have conducted the extensive studies on the cutting machinability of various additive manufactured metal materials [21-23]." maybe changed to "Given the background of the above research, ..."
- Line 76: In this goal of the study, please specify the Al alloys that are used.
- Units: not "2.5mm", "35x9x3mm", "9554~23885 rpm" but "2.5 mm", "(35x9x3)mm", "(9554~23885) rpm"
- Consider putting the sub figures a/, b/, c/ in the same row to use the space more efficiently
- Figure 7, Figure 9: Kindly make the scale more visible 
Content:
- Please specify the AM machine that was used for SLM process, the printing parameters, the orientation of the print, and the printing setup, etc. Since they are important to understand how it relates to the property of the printed samples.
- Table 1: How did the authors obtain the mechanical properties? Was it for horizontal or vertical direction?
- Line 163: "By counting, the average value of the resultant forces is larger about 11.8% with the increase of Si content in additive manufactured AlSi10Mg aluminum alloys.". Kindly support this claim with other literature. Why increase Si makes it larger/better/worse? Is there any published reason from the microstructure point of view?
- Figure 7 Burr morphology characteristics, the machined surface was along the layers or on top of the printed samples?
- Kindly expand the conclusions with more discussion on the AM process and how it affects the machineability of the materials. Comparison with the conventional material can be added as well.

Author Response

Response to Reviewer 4 Comments

We would like to thank the reviewer for giving us the constructive suggestions which would help us to improve the paper quality. Here we submit a new version of our manuscript which has been revised according to the suggestions. All the revisions have been highlighted in the manuscript. We hope that our revised manuscript and response to reviewers submitted here can meet the requirement of the journal Materials. The following is a point-by-point response to the Reviewer 4 comments:

Line 51: "600Mpa" please correct to "600 MPa"

Response: Thank you for the comments. This mistake has been revised.

- Abbreviations: LPBF and SLM are not explained. "Additive manufacturing" is usually abbreviated as "AM". Kindly check the rest.

Response: Thank you for the comments. The explanations for the abbreviations have been added.

Line 62: "Wu et al. have found the significant differences in cutting machinability of additive manufactured titanium alloy compared to the forged titanium alloy, they have pointed out that the machining parameters need to be adjusted for the additive manufactured blank [18]." is a long sentence. Consider breaking it into two.

Response: Thank you for the comments. This long sentence has been broken into two sentences. And we also have carefully checked other sentences in this paper.

- Line 71: "From above, at present, many researchers have conducted the extensive studies on the cutting machinability of various additive manufactured metal materials [21-23]." maybe changed to "Given the background of the above research, ..."

Response: Thank you for the comments. This sentence has been revised based on your suggestions.

Line 76: In this goal of the study, please specify the Al alloys that are used.

Response: Thank you for the comments. The material grade has been added in the goal of this study, based on your suggestions.

Units: not "2.5mm", "35x9x3mm", "9554~23885 rpm" but "2.5 mm", "(35x9x3)mm", "(9554~23885) rpm"

Response: Thank you for the comments. The units in this paper have been revised based on your suggestions.

Consider putting the sub figures a/, b/, c/ in the same row to use the space more efficiently

Response: Thank you for the comments. Based on the paper template of this journal, we have set the separate subheadings.

Figure 7, Figure 9: Kindly make the scale more visible

Response: Thank you for the comments. The scale in the picture has been revised.

Please specify the AM machine that was used for SLM process, the printing parameters, the orientation of the print, and the printing setup, etc. Since they are important to understand how it relates to the property of the printed samples.

Response: Thank you for the comments. In this paper, the AM materials were produced by Falcon Tech Co., Ltd. In SLM, laser power was 340 kW, the scanning speed was 1400 m/s, the scanning interval was 90 μm, and layer thickness was 30 μm. The printing orientation deflects 67° angle from the length direction of workpiece.

Table 1: How did the authors obtain the mechanical properties? Was it for horizontal or vertical direction?

Response: Thank you for the comments. The mechanical properties of the additive manufactured materials were tested from the horizontal direction, which is the actual machined surface.

- Line 163: "By counting, the average value of the resultant forces is larger about 11.8% with the increase of Si content in additive manufactured AlSi10Mg aluminum alloys.". Kindly support this claim with other literature. Why increase Si makes it larger/better/worse? Is there any published reason from the microstructure point of view?

Response: Thank you for the comments. Based on mechanical properties of AlSi10Mg, it presents the restively higher hardness, this can cause the larger cutting force.

- Figure 7 Burr morphology characteristics, the machined surface was along the layers or on top of the printed samples?

Response: Thank you for the comments. The machined surface was on the top of the printed samples.

Kindly expand the conclusions with more discussion on the AM process and how it affects the machineability of the materials. Comparison with the conventional material can be added as well.

Response: Thank you for the comments. Your suggestions have been adopted. In this paper, we mainly focused on the effect of Si contents on machineability, we will further study the machineability by comparing the micro milling machineability between AM materials and casting materials.

Round 2

Reviewer 1 Report

Comments and Suggestions for Authors

Accept in present form

Reviewer 2 Report

Comments and Suggestions for Authors

The manuscript has been partially corrected. The author's responses are adequate but not integrated into the corrected manuscript. However, despite the partial corrections in the manuscript itself, I suggest accepting the manuscript in its current form.

Reviewer 4 Report

Comments and Suggestions for Authors

The authors have carefully addressed all of my comments and have satisfactorily made the necessary revisions. Their changes are convincing and have contributed to the overall improvement of the quality of the article.